# Evaluating the performance of artificial intelligence software for lung nodule detection on chest radiographs in a retrospective real-world UK population

Ahmed Maiter ![ORCID] ,[1,2] Katherine Hocking,[2] Suzanne Matthews,[2,3] Jonathan Taylor,[3] Michael Sharkey,[3] Peter Metherall,[3] Samer Alabed,[1,2] Krit Dwivedi,[1,2] Yousef Shahin,[1,2] Elizabeth Anderson,[2] Sarah Holt,[2] Charlotte Rowbotham,[2] Mohamed A Kamil,[2] Nigel Hoggard,[1,2,4] Saba P Balasubramanian,[3,5] Andrew Swift,[1,2,4] Christopher S Johns ![ORCID] [2]

¹School of Medicine and Population Health, The University of Sheffield, Sheffield, UK
²Radiology, Sheffield Teaching Hospitals NHS Foundation Trust, Sheffield, UK
³Medical Imaging and Medical Physics, Sheffield Teaching Hospitals NHS Foundation Trust, Sheffield, UK
⁴NIHR Sheffield Biomedical Research Centre, Sheffield, UK
⁵Surgical directorate, Sheffield Teaching Hospitals Foundation NHS Trust, Sheffield, UK

**Correspondence to**
Dr Christopher S Johns;
christopher.johns@nhs.net

## ABSTRACT

**Objectives** Early identification of lung cancer on chest radiographs improves patient outcomes. Artificial intelligence (AI) tools may increase diagnostic accuracy and streamline this pathway. This study evaluated the performance of commercially available AI-based software trained to identify cancerous lung nodules on chest radiographs.

**Design** This retrospective study included primary care chest radiographs acquired in a UK centre. The software evaluated each radiograph independently and outputs were compared with two reference standards: (1) the radiologist report and (2) the diagnosis of cancer by multidisciplinary team decision. Failure analysis was performed by interrogating the software marker locations on radiographs.

**Participants** 5722 consecutive chest radiographs were included from 5592 patients (median age 59 years, 53.8% women, 1.6% prevalence of cancer).

**Results** Compared with radiologist reports for nodule detection, the software demonstrated sensitivity 54.5% (95% CI 44.2% to 64.4%), specificity 83.2% (82.2% to 84.1%), positive predictive value (PPV) 5.5% (4.6% to 6.6%) and negative predictive value (NPV) 99.0% (98.8% to 99.2%). Compared with cancer diagnosis, the software demonstrated sensitivity 60.9% (50.1% to 70.9%), specificity 83.3% (82.3% to 84.2%), PPV 5.6% (4.8% to 6.6%) and NPV 99.2% (99.0% to 99.4%). Normal or variant anatomy was misidentified as an abnormality in 69.9% of the 943 false positive cases.

**Conclusions** The software demonstrated considerable underperformance in this real-world patient cohort. Failure analysis suggested a lack of generalisability in the training and testing datasets as a potential factor. The low PPV carries the risk of over-investigation and limits the translation of the software to clinical practice. Our findings highlight the importance of training and testing software in representative datasets, with broader implications for the implementation of AI tools in imaging.

## INTRODUCTION

Early diagnosis of cancer is essential to improve prognosis.[1 2] Lung cancer is the most

## STRENGTHS AND LIMITATIONS OF THIS STUDY

⇒ The artificial intelligence software was tested on chest radiographs from a large real-world cohort of patients with an expected prevalence of lung cancer.
⇒ Performance was assessed against two different reference standards: radiologist detection of nodules and diagnosis of lung cancer by multidisciplinary team (MDT) decision.
⇒ Standard and clinically relevant metrics were used to quantify performance.
⇒ Failure analysis was undertaken to identify factors contributing to incorrect classification of radiographs.
⇒ Limitations include the single-centre retrospective study design, risk of observer bias from manual evaluation of radiograph reports and reliance on documented MDT decision as a reference standard.

common cause of cancer deaths in the UK, with a 10-year survival rate of only 10%.[3] Early lung cancer may be apparent on chest radiographs as a lung nodule, defined as a focal opacity measuring<30 mm.[4] Although the causes of lung nodules are broad, the detection of a new nodule on a chest radiograph should raise suspicion for cancer and usually necessitates further investigation with CT.[5]

In the UK, the National Optimal Lung Cancer Pathway (NOLCP) was developed to expedite diagnosis and treatment to improve lung cancer survival. The pathway goal is to commence treatment within 49 days of referral for patients with a confirmed diagnosis.[6] According to the NOLCP, patients with chest radiograph abnormalities who are suspicious for lung cancer should undergo diagnostic CT, to be performed and reported within 72 hours. This poses a challenge for healthcare services, particularly in the

BMJ

context of increasing demand for medical imaging and staff shortages.[7] The time taken from radiograph acquisition to reporting can delay the pathway and may be problematic in centres with reporting backlogs. Furthermore, chest radiographs are performed in high volumes for a variety of indications, with only a small proportion raising concern for cancer.

The application of artificial intelligence (AI) methods to medical imaging is expanding rapidly and promises a number of benefits.[8–10] Automated evaluation of images could help to triage studies for earlier reporting by radiologists, focusing resource allocation to cases where timely reporting is likely to have the largest clinical benefit. Recent years have seen an increase in the number of studies presenting AI approaches to the evaluation of chest radiographs, either alone or in combination with radiologists. These have included tools to assist with the identification of acute abnormalities such as pneumonia or pneumothorax,[11–14] tuberculosis screening[15–17] and identification of lung cancer.[18–20] Studies have also evaluated how AI-based categorisation of chest radiographs could help to streamline clinical workflows.[21 22] As the field advances and AI tools become commercially available, it is imperative that their performance in real-world populations and clinical settings is evaluated and understood.

The Auto Lung Nodule Detection software (ALND; Samsung Electronics, Suwon, South Korea) was developed to aid identification of lung nodules on digital chest radiographs. The software was class IIa medical devices directive UK Conformity assessed (UKCA) and Conformité Européenne (CE)-marked, Federal Drug Administration (FDA)-approved and commercially available at the time of our study. The design, training and validation of the software have been reported previously.[23 24] The software is based on a deep convolutional neural network algorithm, trained using an enriched dataset of 17 210 radiographs (79.7% normal and 20.3% with cancerous lung nodules) from a single South Korean centre. The software was subsequently tested on 800 radiographs (25.0% normal and 75.0% with cancerous lung nodules) from 4 external datasets (from South Korea, Germany and the USA). The performance when used alone varied between the four enriched datasets, with a sensitivity range of 51.1%–79.1% and false positives per image (FPPI) rate of 0.10–0.30. However, the software has not been tested in an unselected real-world cohort of patients attending for routine chest radiographs.

This study aimed to evaluate the performance of ALND at detecting lung nodules on chest radiographs seen in routine practice in a tertiary UK centre. Radiologist reports and lung cancer multidisciplinary team (MDT) decisions were used as the reference standards for lung nodule and lung cancer diagnosis.

## MATERIALS AND METHODS

This retrospective single-centre study was undertaken at Sheffield Teaching Hospitals NHS Foundation Trust, Sheffield, UK. The study flow is indicated in figure 1. The study is presented in accordance with the Checklist for Artificial Intelligence in Medical Imaging (online supplemental materials).[25] Data analysis was performed using Prism (GraphPad, San Diego, California, USA) and R Studio using the Tidyverse and EpiR packages.

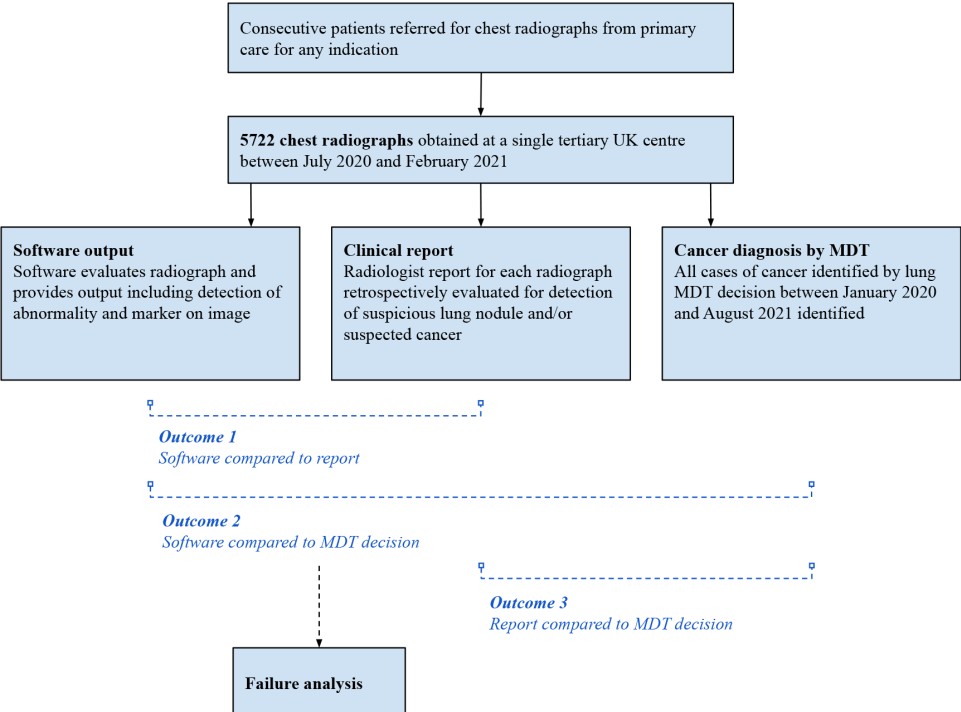

**Figure 1** Overview of study design. No radiographs or patients were excluded. MDT, multidisciplinary team.

## Case selection

Chest radiographs were identified retrospectively from the local radiology information system. Radiographs were eligible for inclusion if they met all of the following criteria: (1) requested by the patient's general practitioner, (2) performed on adult patients (3) between 1 July 2020 and 26 February 2021, and acquired (3) in the posterior-anterior projection, (4) on one of three GC85A digital radiography systems (Samsung Electronics, Suwon, South Korea) running the software tool (ALND V.1.0, Samsung Electronics, Suwon, South Korea) and (5) at the same tertiary centre (Sheffield Teaching Hospitals, UK).

## Evaluation of clinical reports

For each included radiograph, the clinical report was assessed retrospectively for an opinion on the presence of a suspicious nodule or other features of cancer. This was performed manually given the broad variation in reporting styles and language. For example, reports could identify a suspicious nodule explicitly or implicitly (eg, '2-week wait referral is advised' or 'urgent CT is recommended').

## Software results

Included radiographs were assessed by the software on the digital radiography system at the time of acquisition. For each radiograph, the software outputs included the original Digital Imaging and Communications in Medicine (DICOM) formatted image, a binary assessment for the presence of a suspicious lung nodule (ie, 'yes' or 'no'), the number of detected abnormalities and a copy of the image with a marker on each detected abnormality (*not* a contour of the abnormality, but only an indication of location). The outputs were sent to a research-only Picture Archiving Communications System and were not accessible for clinical use.

## Accuracy of nodule detection

For each case, the software output was compared with the corresponding clinical report, which was used as the reference standard for the presence of a suspicious nodule. Sensitivity, specificity, positive predictive value (PPV), negative predictive value (NPV), overall accuracy and FPPI were calculated from contingency tables for all cases. FPPI was calculated as the number of false positive results divided by the number of radiographs.

## Accuracy of cancer diagnosis

Records from the local lung cancer MDT meetings were reviewed from the date of the first included radiograph to 6 months after the last included radiograph. Cases with a diagnosis of cancer by lung MDT decision were identified. Diagnosis of lung cancer by MDT opinion was used as the reference standard, against which the software output and clinical report were compared for each case. Sensitivity, specificity, PPV, NPV, overall accuracy and FPPI were calculated from contingency tables. In those cases where a cancer was identified by the software, the output was interrogated to ensure that the identified abnormality

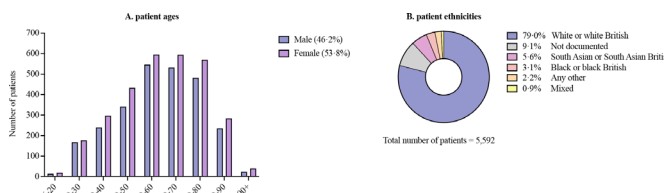

**Figure 2** (A) Number of patients by age group and sex. (B) Documented patient ethnicities.

was in the correct location; if the marker had been placed incorrectly by the software this was counted as a negative result. For each case with a diagnosis of cancer, the radiograph was re-reviewed by one consultant thoracic radiologist (CSJ, 11 years of experience) to determine if the abnormality was visible; subgroup analysis was performed excluding cases in which the cancerous abnormality was not visible on the radiograph.

## Failure analysis of incorrectly classified cases

Failure analysis was performed for false positive and false negative outputs by the software in reference to the presence or absence of MDT-confirmed cancer. For each of these cases, the software outputs were manually compared with the radiograph, clinical report and MDT opinion.

## Patient and public involvement

There was no patient or public involvement in this study.

# RESULTS

## Included radiographs and patients

A total of 5722 chest radiographs were included from 1 July 2020 to 26 February 2021. These were acquired for 5592 different patients, with 2.3% of these having had either 2 or 3 radiographs. The included patients had a median age of 59 years (IQR 46–72 years), with 53.8% female (figure 2A). Overall, 79.0% of patients were white or white British. Ethnicity information was unavailable for 9.1% of cases (figure 2B). No radiographs or patients were excluded.

## Clinical reports

All radiographs had been reported independently by FRCR-qualified radiologists, with 91.0% reported by seven different consultant thoracic radiologists. Overall, 58.5% of reports mentioned comparison with previous imaging studies. Potentially cancerous abnormalities were identified in 3.0% of reports; these included nodules (59.1%), masses (25.7%) and other abnormalities such as lung volume loss or pleural thickening (21.6%).

## Accuracy of nodule detection

All 5722 chest radiographs were evaluated by the software, with none failing evaluation. The software identified a total of 1120 potentially cancerous nodules on 17.5% of radiographs. When compared against identification of suspicious nodules in the clinical report (ie, excluding masses or other abnormalities for which the software had

**Table 1** Performance of the software and clinical reports against reference standards, with 95% CI

| Test | Suspicious nodule identified by software | Suspicious nodule identified by software | Any suspicious abnormality on clinical report* |
|---|---|---|---|
| Reference standard | Suspicious nodule on clinical report† | Cancer diagnosis by multidisciplinary team decision | |
| Sensitivity | 54.5% (44.2% to 64.4%) | 60.9% (50.1% to 70.9%) | 66.3% (55.7% to 75.8%) |
| Specificity | 83.2% (82.2% to 84.1%) | 83.3% (82.3% to 84.2%) | 98.0% (97.7% to 98.4%) |
| PPV | 5.5% (4.6% to 6.6%) | 5.6% (4.8% to 6.6%) | 35.7% (30.5% to 41.2%) |
| NPV | 99.0% (98.8% to 99.2%) | 99.2% (99.0% to 99.4%) | 99.4% (99.3% to 99.6%) |
| Accuracy | 82.7% (81.7% to 83.6%) | 82.9% (81.9% to 83.9%) | 97.5% (97.1% to 97.9%) |
| FPPI | 0.18 | 0.18 | 0.02 |

*Includes nodules, masses and secondary features of malignancy (such as unilateral pleural effusions or lobar collapse).
†Other suspicious abnormalities (including masses and secondary features of cancer, such as lung volume loss) were excluded.
FPPI, false positives per image; NPV, negative predictive value; PPV, positive predictive value.

not been trained), the software yielded a sensitivity of 54.5%, specificity of 83.2%, PPV of 5.5%, NPV of 99.0% and overall accuracy of 82.7%, with an FPPI of 0.18 (table 1 and online supplemental table 1A).

### Accuracy of cancer diagnosis
A total of 92 patients (1.6%) were found to have a diagnosis of cancer by MDT opinion. When the suspicious nodules identified by the software were compared against MDT diagnosis, the software yielded a sensitivity of 60.9%, specificity of 83.3%, PPV of 5.6%, NPV of 99.2% and accuracy of 82.9%, with an FPPI of 0.18 (table 1 and online supplemental table 1B). Examples of correct identification of cancer by the software are shown in figure 3. When all suspicious abnormalities identified in the clinical reports (ie, including nodules, masses and other abnormalities) were compared against MDT diagnosis, the clinical reports yielded a sensitivity of 66.3%, specificity of 98.0%, PPV of 35.7%, NPV of 99.4% and accuracy of 97.5%, with an FPPI of 0.02 (table 1 and online supplemental table 1C). Hypothetically, if the detection of an abnormality automatically resulted in patients undergoing CT, the software would have resulted in 999 scans with 60.9% of

cancer cases detected (online supplemental table 1B). By comparison, the clinical reports would have resulted in 171 scans with 66.3% of cancer cases detected (online supplemental table 1C).

Of the 92 cases of cancer, the abnormality was deemed to be visible on the radiograph in 75 cases. Using these cases as the reference standard, the software yielded a sensitivity of 68.0%, specificity of 83.3%, PPV of 5.1%, NPV of 99.5% and accuracy of 83.0% (online supplemental table 1D) and online supplemental table 2). The clinical reports yielded a sensitivity of 80.0%, specificity of 98.0%, PPV of 35.3%, NPV of 99.7% and accuracy of 97.8%. (online supplemental table 1E and online supplemental table 2). Performance of the clinical reports was similar to expected.[26]

### Failure analysis of ALND results
The software yielded 34 false negatives compared with MDT opinion (36.9% of cases with cancer). In 22 of these cases, the abnormalities were found to be visible on the radiograph and included masses (63.5 %, defined as >30 mm), lung collapse secondary to obstructing cancer (18.2 %), solitary or multiple cancerous nodules (13.6%), malignant pleural effusion (9.1%) and persistent consolidation (9.1%) (figure 4).

The software yielded 943 false positive results in comparison with MDT opinion, representing 16.8% of all cases without cancer. Only 7.8% of these cases had been reported as abnormal in the formal clinical report and were subsequently found to be due to a benign cause. Contrastingly, the formal clinical report yielded 110 false positive results compared with MDT opinion, representing 2.0% of cases without cancer. Causes of failure by the software included normal anatomy (69.9%), non-cancerous pathology (31.2%) and technical factors (4.8%) (figure 5 and online supplemental table 3).

### DISCUSSION
The evaluation of AI tools in real-world settings is essential for their translation to clinical practice. In this

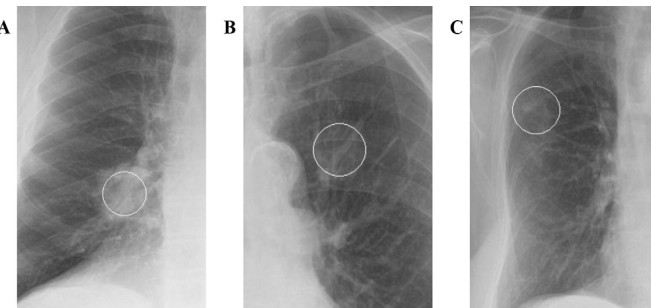

**Figure 3** Examples of true positive cancerous nodule identification by the software. The circles represent markers placed at the location of the detected abnormality by the artificial intelligence (AI) tool (note that these are not contours of the abnormality). (A) Primary right middle lobe lung cancer. (B) Primary left upper lobe lung cancer. (C) Right upper lobe lung metastasis.

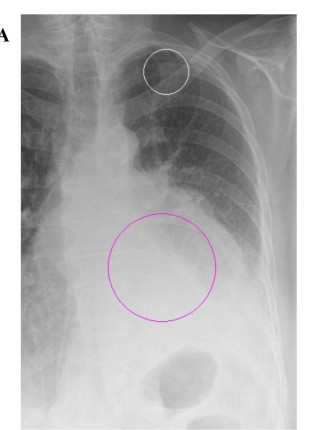
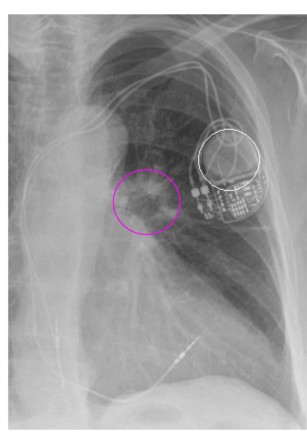

**Figure 4** Examples of false negative results. The white circles represent markers placed at the location of the detected abnormality by the artificial intelligence tool (note that these are not contours of the abnormality). The magenta circles have been added manually to indicate the location of the missed true abnormalities. (A) Missed left lower lobe cancer (magenta circle); the software has also misidentified the left first rib as a false positive abnormality. (B) Missed cancerous left hilar nodule; the software has misidentified the pacemaker as a false positive abnormality.

retrospective study, we evaluated the performance of an AI-based software tool trained to identify suspicious lung nodules on chest radiographs. The software demonstrated underperformance on consecutive patients referred from primary care. While sensitivity and FPPI values were similar to those reported previously for enriched datasets, we have demonstrated a considerably low PPV when tested in a real-world dataset. The results carry important

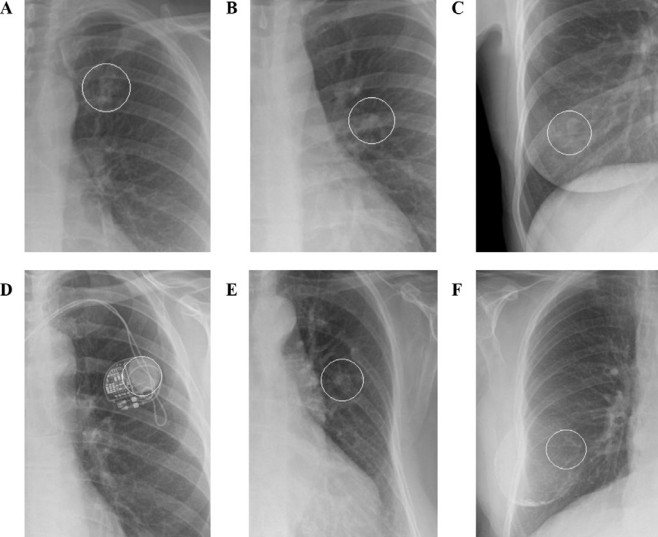

**Figure 5** Examples of false positive abnormalities detected by the software. The circles represent markers placed at the location of the detected abnormality by the artificial intelligence tool (note that these are not contours of the abnormality). (A) End of the left first rib. (B) Composite left perihilar shadows. (C) Right nipple shadow. (D) Pacemaker. (E) Old left rib fracture. (F) Right breast implant.

implications for the training, evaluation and clinical implementation of AI tools in medical imaging.

A large number of consecutive cases referred from primary care were included using broad inclusion criteria. We believe that this unenriched dataset is more representative of the population encountered in routine clinical practice. The software demonstrated low PPV values, which severely limit the potential utility of the software and make it unsuitable for our population. Compared with radiologist reports, the software underperformed in all measures of diagnostic performance. Our results raise important questions about the intended use of the software and how it could fit into a clinical pathway. The low PPV demonstrated here suggests that use of the software for its intended purpose would result in over-investigation of patients without an improvement in sensitivity. In comparison with the clinical reports, if all cases where the software had detected an abnormality suspicious for cancer had undergone CT, this would have resulted in 5.8 times the number of scans performed and fewer cases of cancer detected. Such over-investigation is harmful: unnecessary diagnostic tests carry physical risks such as increased doses of ionising radiation, may contribute to patient anxiety and distress and can increase the demands on healthcare systems. Our results highlight the difference between AI *model* performance and *clinical* performance. The software yielded similar sensitivity and FPPI results to those during its previous validation and testing in enriched datasets,[23 24] but showed very low PPV when applied to a real-world population with normal disease prevalence, representing underperformance for its intended clinical purpose.

Understanding *why* underperformance has occurred is essential for further development of AI tools. As part of the output for each radiograph, the software labelled detected abnormalities. This is an important design feature, enabling failure analysis, allowing proper evaluation of performance and aiding decision-making if used in clinical settings. The software frequently misidentified normal, variant or age-related anatomy as abnormalities. This included false positives that radiologists readily dismiss, such as the ends of the first ribs. What factors could have contributed to this? One potential explanation is the use of insufficiently representative and generalisable datasets for software training and testing. The training and testing datasets were enriched and included a prevalence of cancer far higher than that seen in real world clinical populations, which may have skewed performance towards overcalling abnormalities (https://www.zotero.org/google-docs/?fmenDL).[20 23] In contrast, our dataset had a more realistic prevalence of 1.6%. Differences in population demographics may have also contributed to underperformance. The software was trained on a South Korean population.[23] Gichoya *et al* recently demonstrated that AI tools are capable of determining a patient's race from radiographs, highlighting that there can be an inherent difference in normal imaging appearances between populations.[27 28] An AI model

trained on data from one population may not generalise well to others that show significant demographic differences. The interaction of AI with patient factors such as sex, race and ethnicity is an important practical and ethical issue for the field moving forwards. Alternatively, insufficient variation and volume in the 'normal' radiographs used in training may have affected performance. The spectrum of normality is broad and can encompass variations in anatomy and technical factors relating to radiograph acquisition. How 'normal' radiographs were defined in the training and testing datasets was not reported clearly. For example, did these include anatomical variants, age-related changes, radiographs that were rotated or artefacts from external clothing—all of which are seen commonly in routine practice? And if so, were these included in a sufficient and representative number? Furthermore, the training and testing datasets did not include non-cancerous pathology. For example, smoking and industrial exposures are risk factors for lung cancer and cause benign abnormalities that are visible on chest radiographs, such as emphysema or pleural plaques. It is important for an AI tool to be able to identify cancer in a population of patients with co-existing pathologies that reflect an increased cancer risk. Masses and secondary features of cancer (such as pleural effusion or mediastinal lymphadenopathy) were also absent from the training and testing datasets. Finally, the software was trained to assess radiographs in isolation, with no comparison to previous imaging or clinical context, in stark contrast to how radiologists interpret imaging. In short, training and testing datasets need to be representative of the intended population to ensure generalisability of AI tools, and models should ideally integrate information from previous imaging studies, patient demographics and clinical history. While existing studies have often used enriched data sets to develop and test models, promising models should be tested in representative real-world datasets and environments prior to their routine clinical use.

Our study has limitations. The retrospective design enabled a large study population but did not afford the same degree of robustness or control over confounding variables as a prospective study. Using data from routine clinical care risks incomplete information, such as a lack of documented ethnicity in some patients. There is also inherent difficulty in defining the reference standard for a suspicious nodule. Other studies have used CT or biopsy results, but such approaches are less practical when evaluating a large real-world clinical cohort and introduce their own selection bias: some inflammatory nodules will be transient and not warrant further investigation, some patients may not undergo CT or biopsy, or some results may be indeterminate. We used cancers diagnosed by lung MDT decision as a reference standard, but some cases of cancer may have been missed if they had not been referred to or reviewed by the MDT, or if they represented metastatic disease to the lung from another known primary cancer. Finally, assessment of the clinical reports and failure analysis were performed by radiologists, with the risk of interobserver variability or observer bias. Regardless, we believe that the study limitations are unlikely to explain the considerable differences in performance compared with the software's prior validation and testing. Looking ahead, it is important that software tools showing promising performance in representative datasets such as ours are then more formally evaluated prospectively, such as through randomised controlled trials.

The excitement surrounding AI is understandable given the promise of automation in an era of growing demands for medical imaging. The Royal College of Radiologists reports a shortfall of 29% of the UK radiologist workforce, raising concerns about reporting backlogs and the potential for delayed diagnoses.[7] Chest radiographs remain a mainstay of imaging in radiology departments, with a median of just under 650 000 performed per month in England in 2021–2022.[29] The ability to streamline reporting would assist in dealing with this high burden, addressing backlogs and minimising the impacts of service pressures. Early identification of patients with possible lung cancer on chest radiographs would facilitate more efficient workflows, reducing the time to diagnosis and improving patient outcomes.[1 2] However, this study highlights fundamental questions about the application of AI in medical imaging. First, what constitutes sufficient training and testing? We have demonstrated that performance in narrow and enriched datasets during software development is not necessarily generalisable to real-world populations. This issue is likely to become more significant as the field advances and model performances improve.[21] However, when the pretest probability of the disease is low, large population samples will be required to demonstrate significant differences in performance. There remain significant ethical concerns about the systematic propagation of human bias in AI tools that may disadvantage minority populations and datasets need to be selected carefully and presented in a transparent manner.[27 28 30 31] Second, what are the appropriate benchmarks for performance? Proper definition of reference standards can be challenging, particularly in clinical settings. Radiologist opinion is a common reference standard for medical imaging AI tools and is an intuitive benchmark when considering the utility of a tool, but is subject to its own internal error rate and both intraobserver and interobserver variability. In this study, we have not only compared the software performance against radiologist reports, but also MDT diagnosis of lung cancer, which accounted for clinical, imaging and histopathological factors. The use of MDT diagnosis as a reference standard reflects real-life clinical practice and is an appropriate benchmark when considering the clinical utility of AI tools for cancer detection. Third, how should performance be presented and interpreted? High-quality reporting of model design, training, validation and testing is crucial for understanding the performance and generalisability of AI tools.[32] Transparency of AI tool development and performance is essential for the

gaining the trust of stakeholders, including the public, and is therefore important for translation of tools into the clinical sphere. The way AI tools are presented should be consistent, enabling direct comparisons of performance, and accessible to all stakeholders, allowing purpose and performance to be understood by those without extensive experience in the field of AI. Poor performance should also be presented, analysed and published in a fair manner to minimise bias within the field. Last, where do we see such AI tools fitting into clinical pathways? We found a low PPV, which combined with a low prevalence of disease in the studied population, suggests that implementation of the software would result in over-investigation and potentially *contributing* to the burden on healthcare systems. Conversely, the NPV of the software exceeded 99%, and there could be potential utility in using the software to streamline reporting by removing studies that do not have cancer. Other studies have presented the use of AI approaches to categorise chest radiographs for the purpose of report prioritisation, which is of particular appeal given the large volumes requiring reporting.[21 22] However, in this case the software would also exclude non-cancerous abnormalities that could still be significant (such as infection) and would offer no improvement over a radiologist's NPV, and therefore may not add much clinical value. Alternatively, AI tools may be of benefit when used to augment reporting by radiologists or to provide a second reading.[13 33 34] In order to streamline workflows and pathways, performance thresholds need to be chosen carefully and determined prior to software development.

## CONCLUSION

Evaluation of AI-based software tools in representative patient cohorts is crucial for their translation to routine clinical practice. We have demonstrated underperformance of the software using a large retrospective and relatively unselected clinical cohort, despite apparent similar diagnostic accuracy in comparison to the initial training and testing. Understanding the external validity of software performance is an important consideration as the field continues to advance.

**Contributors** CSJ, SM, PM and JT conceptualised and designed the study. PM developed software for data curation and analysis. CR, SH and CSJ undertook data curation. KH, SH, YS and EA extracted information from the dataset. CSJ, SM and KH performed image failure analysis. CSJ and AM performed formal data analysis for all data. AM and CSJ wrote the original manuscript, which was edited and reviewed by KH, AS, SA, NH, SPB, CR and MAK. AM and CSJ produced the final manuscript. All authors have read and approved the final version of the manuscript. AM, CSJ and KH had direct access to and verified the data presented in the manuscript. AM and CSJ revised the manuscript following peer review. CSJ is the guarantor.

**Funding** Samsung Electronics (Suwon, South Korea) provided free access to the Auto Lung Nodule Detection (ALND) software for the Sheffield Teaching Hospitals for the purposes of this study. AJS is supported by a Wellcome Trust Fellowship Grant (205188/Z/16/Z). KD is supported by a Wellcome Trust Fellowship Grant (222930/Z/21/Z and 4ward North 203914/Z/16/). This research was carried out in part at the National Institute for Health and Care Research (NIHR) Sheffield Biomedical Research Centre (BRC).

**Disclaimer** The views expressed are those of the author(s) and not necessarily those of the funders. For the purpose of open access, the author has applied a CC BY public copyright licence to any author accepted manuscript version arising from this submission.

**Competing interests** None declared.

**Patient and public involvement** Patients and/or the public were not involved in the design, or conduct, or reporting, or dissemination plans of this research.

**Patient consent for publication** Not applicable.

**Ethics approval** Study approval was obtained from the local clinical research office, a local ethics committee (Sheffield 3DLab; reference 17/YH/0142), and information governance team. Requirement for direct patient consent was waived by the clinical research office and ethics committee.

**Provenance and peer review** Not commissioned; externally peer reviewed.

**Data availability statement** Data are available upon reasonable request. Anonymised participant data can be provided on reasonable request pending local information governance approval, which may require a signed data access agreement.

**ORCID iDs**
Ahmed Maiter http://orcid.org/0000-0002-4999-2608
Christopher S Johns http://orcid.org/0000-0003-3724-0430

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
