## [Reviewer comments · BMJ Open]

ARTICLE DETAILS

TITLE (PROVISIONAL)	Evaluating the performance of artificial intelligence software for lung nodule detection on chest radiographs in a retrospective real-world UK population.
AUTHORS	Maiter, Ahmed; Hocking, Katherine; Matthews, Suzanne; Taylor, Jonathan; Sharkey, Michael; Metherall, Peter; Alabed, Samer; Dwivedi, Krit; Shahin, Yousef; Anderson, Elizabeth; Holt, Sarah; Rowbotham, Charlotte; Kamil, Mohamed; Hoggard, N; Balasubramanian, Saba; Swift, Andrew; Johns, Christopher

VERSION 1 – REVIEW

REVIEWER	Lim, Gilbert National University of Singapore, School of Computing
REVIEW RETURNED	30-Aug-2023

GENERAL COMMENTS	We thank the authors for largely addressing our previous comments. A couple of minor points remain. 1. The relevancy/significance of the two reference standards (i.e. radiologist report, MDT cancer diagnosis) might be explained further. Are both reference standards of practical clinical utility? Moreover, a brief analysis of these two reference standards against each other might be considered if possible, to provide additional context.2. The relative performance as achieved in this study, as compared to the original external validation performance appears to be broadly comparable (Table 1), in the sense that both the sensitivity and FPPI values are within the original external validation range. The summary description of the ALND software as "underperformance" might thus be further justified (e.g. to existing clinical standards)
--

REVIEWER	Jha, Saurabh University of Pennsylvania
REVIEW RETURNED	12-Sep-2023

GENERAL COMMENTS	The authors have answered my concerns about the paper. The paper would benefit from an accompanying editorial.
--

VERSION 1 – AUTHOR RESPONSE

Reviewer: 1

Dr. Gilbert Lim, National University of Singapore

Comments to the Author:

We thank the authors for largely addressing our previous comments. A couple of minor points remain.

We thank Dr Lim for their review and pertinent comments.

1. The relevancy/significance of the two reference standards (i.e. radiologist report, MDT cancer diagnosis) might be explained further. Are both reference standards of practical clinical utility? Moreover, a brief analysis of these two reference standards against each other might be considered if possible, to provide additional context.

As touched upon in our discussion of the study limitations, choosing appropriate reference standards for benchmarking AI tools is not always straightforward. We believe that the two reference standards used here are complementary and a strength of this study.

Expert radiologist opinion is a very common reference standard for diagnostic medical imaging AI tools and is an important benchmark when considering whether an AI tool can be implemented clinically ('is the AI software as good as the person currently carrying out the task?'). This is often a feasible comparison and one we have made by comparing the software outputs against the formal clinical reports for the radiographs.

However, radiologists sometimes disagree and make errors, so this reference standard has flaws. What is more challenging is determining whether an AI tool is actually detecting cancer. There are various reference standards that could be used (e.g. serial follow up CT, biopsy results). Our use of an MDT cancer diagnosis reflects real-life practice. MDT diagnosis involves expert opinion and integrates clinical history with imaging and histopathology. While it does have flaws, it's what we currently use for cancer diagnosis in real-life practice and therefore made a useful reference standard.

We have reworded some of the discussion to explore these points.

2. The relative performance as achieved in this study, as compared to the original external validation performance appears to be broadly comparable (Table 1), in the sense that both the sensitivity and FPPI values are within the original external validation range. The summary description of the ALND software as "underperformance" might thus be further justified (e.g. to existing clinical standards)

We agree that the software's sensitivity and FPPI values are comparable to those reported in the previous external validation publication. However, we believe that our study highlights the differences between model performance and clinical performance.

The model performance was as previously published. However, the model was trained and tested using enriched datasets and when applied to our real-world patient population showed too low a PPV to make it of clinical utility. We highlight that, based on this cohort, use of the software for its intended purpose may have resulted in around 6 times the number of CT scans being done for no improvement in the number of cancers detected. The "underperformance" here is clinical and refers to the intended use of the software.

We have reworded some of the discussion to emphasise this distinction.

Reviewer: 2

Saurabh Jha, University of Pennsylvania

Comments to the Author:

The authors have answered my concerns about the paper. The paper would benefit from an accompanying editorial.

We thank Dr Jha for their review and suggestion for an accompanying editorial.